# Molecular design strategy of fluorogenic probes based on quantum chemical prediction of intramolecular spirocyclization

Ryo Tachibana[1], Mako Kamiya[2,3], Satoshi Suzuki[4], Keiji Morokuma[4], Aika Nanjo[1] & Yasuteru Urano [1,2,5✉]

Fluorogenic probes are essential tools for real-time visualization of dynamic intracellular processes in living cells, but so far, their design has been largely dependent on trial-and-error methods. Here we propose a quantum chemical calculation-based method for rational prediction of the fluorescence properties of hydroxymethyl rhodamine (HMR)-based fluorogenic probes. Our computational analysis of the intramolecular spirocyclization reaction, which switches the fluorescence properties of HMR derivatives, reveals that consideration of the explicit water molecules is essential for accurate estimation of the free energy difference between the open (fluorescent) and closed (non-fluorescent) forms. We show that this approach can predict the open-closed equilibrium ($pK_{cycl}$ values) of unknown HMR derivatives in aqueous media. We validate this $pK_{cycl}$ prediction methodology by designing red and yellow fluorogenic peptidase probes that are highly activated by γ-glutamyltranspeptidase, without the need for prior synthesis of multiple candidates.

[1] Graduate School of Pharmaceutical Sciences, The University of Tokyo, 7-3-1 Hongo, Bunkyo-ku, Tokyo 113-0033, Japan. [2] Graduate School of Medicine, The University of Tokyo, 7-3-1, Hongo, Bunkyo-ku, Tokyo 113-0033, Japan. [3] PRESTO, Japan Science and Technology Agency, 4-1-8 Honcho, Kawaguchi, Saitama 332-0012, Japan. [4] Fukui Institute for Fundamental Chemistry, Kyoto University, Takano-Nishibiraki-cho 34-4, Sakyou-ku, Kyoto 606-8103, Japan. [5] AMED CREST, Japan Agency for Medical Research and Development, 1-7-1 Otemachi, Chiyoda-ku, Tokyo 100-0004, Japan. ✉email: uranokun@m.u-tokyo.ac.jp

Fluorogenic probes play a fundamental role in real-time imaging of a variety of dynamic intracellular processes. In order to develop fluorogenic probes with high sensitivity, it is important to precisely control the fluorescence properties before and after reaction/interaction with the target molecules. Several mechanisms are used in the design of fluorogenic probes, including photoinduced electron transfer (PeT)[1], Förster resonance energy transfer (FRET)[2], intramolecular spirocyclization, and intramolecular charge transfer (ICT)[3]. Among these mechanisms, the rate of PeT can be predicted by the Rehm–Weller equation[4] and the Marcus' theory of electron transfer reactions[5,6], and that of FRET can be predicted by the Förster equation[7,8], thus providing, in principle, a rational basis for probe design.

In contrast to PeT and FRET, which are deactivation processes from the excited state, intramolecular spirocyclization is a ground-state equilibrium between a colorless/nonfluorescent spirocyclic form and a colored/fluorescent form. Since this equilibrium enables complete quenching by breaking the π-conjugation of fluorophore scaffold, probes based on intramolecular spirocyclization can exhibit significant fluorescent activation when the equilibrium of the two forms is appropriately shifted. For example, fluorescein diacetate (FDA) probe for intracellular esterase[9] or rhodamine spiroamide-based probes for metal ions[10] exist in the colorless and nonfluorescent spirolactone or spiroamide form in the absence of their targets, but are converted to the colored and fluorescent xanthene form upon reaction with the targets. More recently, we have expanded the design strategy based on intramolecular spirocyclization, by changing the intramolecular nucleophile at position-2′ of rhodamine derivatives from carboxylate or amide to a more nucleophilic group such as hydroxymethyl, aminomethyl, or mercaptomethyl[11], and this approach has enabled us to develop new fluorogenic probes for hypochlorous acid[11], oxidoreductase[12], aminopeptidases[13,14], pH-activatable probes[15], and super-resolution imaging[16,17]. For example, we utilized the distinctive spirocyclic nature of hydroxymethyl rhodamine green (HMRG) derivatives to design and develop highly sensitive fluorogenic probes for aminopeptidases overexpressed in cancer cells[13,18,19]. These probes exist in colorless and nonfluorescent spirocyclic form at the physiological pH of 7.4, but are converted to HMRG, which exists in the fluorescent xanthene form, upon one-step hydrolysis by the target enzymatic activity, resulting in rapid and dramatic fluorescence activation. These probes enabled not only in vivo imaging of cancer in mouse models, but also ex vivo fluorescence imaging of cancers in freshly resected specimens from patients.

However, in spite of the usefulness of fluorogenic probes based on intramolecular spirocyclization, rational design is still difficult, due to the lack of a method to predict the equilibrium constant of intramolecular spirocyclization of probe candidates prior to synthesis. Considering that intramolecular spirocyclization is a ground-state equilibrium, we thought that it should be possible to predict the equilibrium constant with high accuracy by means of computational chemistry. Such methodology would have the potential to revolutionize the design of spirocyclization-based probes by minimizing or eliminating the need for time-consuming synthesis of multiple candidates.

In this paper, we propose a quantum chemical calculation-based method to predict the equilibrium constant of intramolecular spirocyclization of hydroxymethyl rhodamine (HMR) derivatives. As an indicator of the equilibrium constant of intramolecular spirocyclization, we focused on the pH-dependence of the equilibrium of the two forms and used the $pK_{cycl}$ value, which we have defined as the pH value at which the absorbance/fluorescence derived from the open form is half of the maximum (Fig. 1). Since we have already determined the $pK_{cycl}$ values of several HMR derivatives, we first aim at exploring the quantum chemical prediction of the $pK_{cycl}$ values of these HMR derivatives. Then, we use the developed methodology to design red and yellow fluorogenic peptidase probes that are highly activated by γ-glutamyltranspeptidase, and validate it by synthesizing the designed compounds, measuring their $pK_{cycl}$ values, and confirming their practical utility.

## Results and discussion

**Calculation of intramolecular spirocyclization.** We first examined the correlation between $pK_{cycl}$ values and parameters that can be easily obtained by structural optimization using quantum chemistry calculations, such as the C–O bond length of the spiro-ring and the lowest unoccupied molecular orbital (LUMO) energy level of fluorophore. The former would be related to the stability of the spiro ring and the latter would reflect the electrophilicity of the fluorophore. However, there was no correlation between $pK_{cycl}$ and these values (Supplementary Figs. 1 and 2), so we decided to perform more detailed calculations to predict the $pK_{cycl}$ values of HMR derivatives.

As shown in Fig. 1, the intramolecular equilibrium of HMR derivatives consists of an acid–base equilibrium of the amino group of the xanthene ring and the hydroxymethyl group of the benzene ring, and two types of spiro-ring-opening and -closing reactions. The hydroxymethyl group of the benzene ring works as an intramolecular nucleophile, attacking the carbon atom at position-9 of the xanthene fluorophore to form a closed spirocycle. Open forms under acidic and basic conditions ($O_A$ and $O_B$, respectively) have strong absorption and fluorescence emission in the visible wavelength region derived from the extended π-conjugation of the xanthene fluorophore, while closed forms under acidic and basic conditions ($C_A$ and $C_B$, respectively) have no absorption or fluorescence emission in the visible wavelength region because of the deconjugation of the xanthene fluorophore.

Assuming that only these four species are involved in the equilibrium, $pK_{cycl}$ can be interpreted as the pH at which the concentration of ring-opened forms ($O_A + O_B$) is equal to that of ring-closed forms ($C_A + C_B$). Then, $pK_{cycl}$ can be expressed as equation (1) in Fig. 2 by using the equilibrium constants $K_{aOH}$ ($K_a$ of benzyl alcohol), $K_{aNH}$ ($K_a$ of anilines), and $K_A$ (open–closed equilibrium under acidic conditions). In this equation, $K_{aOH}$ and $K_{aNH}$ can be replaced with the reported $pK_a$ values of similar structures[20–22] (benzyl alcohol, aniline, N,N-dimethylaniline) ($K_{aOH} = 10^{-15.4}$, $K_{aNH} = 10^{-4.6}$(NR$_2$ = NH$_2$), $10^{-4.9}$(NR$_2$ = di or monoalkylated amine). In the case of derivatives that have an aminomethyl group instead of a hydroxymethyl group, such as AMRG, we assumed that the concentration of $O_B$ can be ignored ($K_{aOH} = 0$), because the amino group is hardly deprotonated under aqueous conditions. Therefore, $pK_{cycl}$ can be predicted if $K_A$ can be accurately estimated. $K_A$ can be expressed as equation (2) in Fig. 2, in which ΔG represents the difference in free energy between the open form and closed form under acidic conditions ($O_A$ and $C_A$, respectively). Based on these considerations, we decided to evaluate the difference in free energy between $O_A$ and $C_A$ (ΔG) by means of quantum chemical calculation to accurately predict $pK_{cycl}$ values.

In order to calculate the free energy difference between $O_A$ and $C_A$, we carefully handled the effect of water molecules of the solvent around the HMR derivatives, since in our previous studies we found that HMR derivatives show dramatic spirocyclization equilibrium changes only in protic solvents, such as aqueous buffer. In fact, when we calculated the free energy using only the dielectric field approximation without considering the direct effect of hydrogen bonding with water, the open form was predicted to

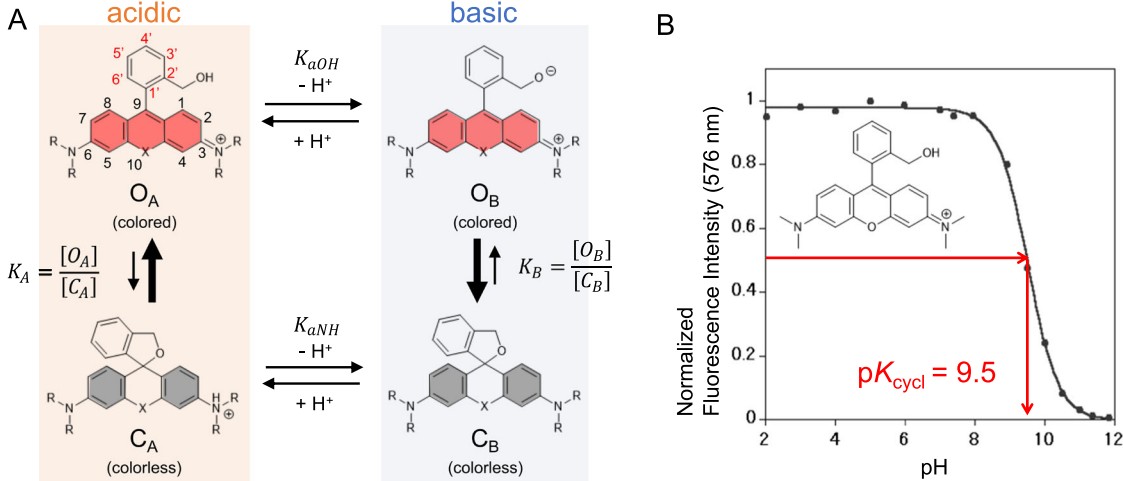

**Fig. 1 Intramolecular spirocyclization of HMR derivatives. a** Acid–base equilibrium of HMR derivatives. **b** Correlation between pH and normalized fluorescence intensity of HMTMR.

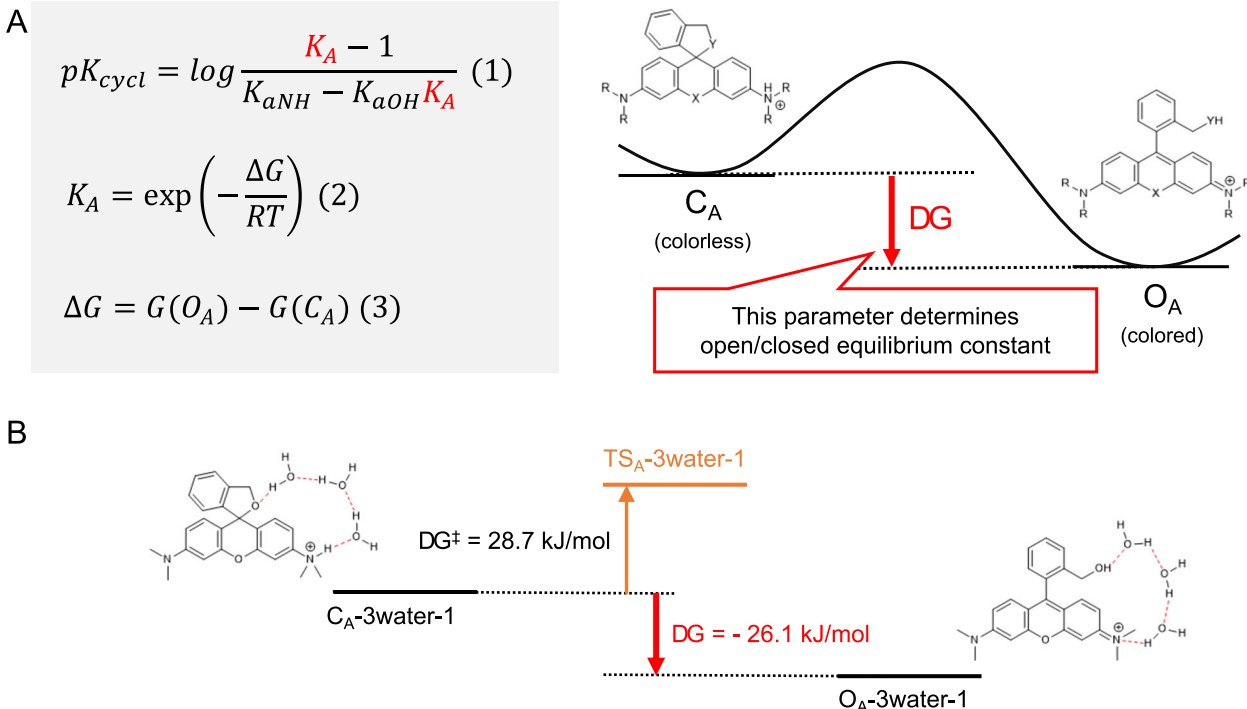

**Fig. 2 Calculation of intramolecular spirocyclization in acidic condition. a** Formula for $pK_{cycl}$ based on statistical mechanics. **b** Calculated free energy difference between open/closed form and activation free energy of HMTMR with a 3-water bridge.

be much more dominant than is actually the case, resulting in no correlation with $pK_{cycl}$ (Supplementary Table 1 and Supplementary Fig. 3). We speculated that the positive charge localized on aniline in the closed form would be stabilized by hydrogen bonds from surrounding water molecules. Therefore, we decided to include water molecules interacting directly with HMR derivatives through hydrogen bonds (first-shell water) in the calculation of free energy difference in order to see whether the free energy differences calculated in this way would reproduce the experimentally obtained values. First, we placed a water molecule adjacent to the cationic proton of the closed form to take the cation-delocalization effect into account, and also adjacent to the hydroxymethyl group of the open form to match the number of hydrogen bonds included in the calculations. We calculated $pK_{cycl}$

again for these structures (Supplementary Fig. 4). The result was improved, but still not sufficiently accurate. Based on this finding, we expected that the calculated $pK_{cycl}$ would converge to an accurate value if we added water molecules one by one for energy evaluation. We tested various positions of a second first-shell water and found that the stabilization of closed form was greatest when two water molecules are connected in series (structure 1,5 in Supplementary Fig. 5). We calculated $pK_{cycl}$ for these structures (Supplementary Figs. 6 and 7) and found that the result was further improved. Next, we tested various structures with three first-shell waters based on the stable structures with two first-shell waters, and found that the structure was most stable when the amino group of the xanthene ring and the hydroxymethyl group of the benzene ring were linked via three water molecules

**Table 1 Comparison between measured and calculated p$K_{cycl}$ values of HMR derivatives.**

| | X | Y | Z | R$^1$ | R$^2$ | Measured p$K_{cycl}$ | Calculated p$K_{cycl}$ | Error |
|---|---|---|---|---|---|---|---|---|
| HMRG | O | O | H | H | H | 8.1 | 11.3 (7.9[a]) | 0.2 |
| AMRG | O | NH | H | H | H | 6.2 | 10.1 (6.2[a]) | 0.0 |
| HMTMR | O | O | H | Me | Me | 9.5 | 9.5 | 0.0 |
| AMTMR | O | NH | H | Me | Me | 7.8 | 8.1 | 0.3 |
| HMRB | O | O | H | Et | Et | 9.2 | 9.3 | 0.1 |
| AMRB | O | NH | H | Et | Et | 8.2 | 8.1 | 0.1 |
| HMSiR | SiMe$_2$ | O | H | Me | Me | 5.7 | 6.2 | 0.5 |
| AMSiR | SiMe$_2$ | NH | H | Me | Me | 4.2 | 4.8 | 0.6 |
| HMDiMeR | O | O | H | H | Me | 8.9 | 9.2 | 0.3 |
| HMDiEtR | O | O | H | H | Et | 9.3 | 8.8 | 0.5 |
| HMJR | O | O | H | H | Julolidine[b] | 10.3 | 10.7 | 0.4 |
| HMDiMeFR | O | O | F | H | Me | 8.2 | 8.3 | 0.1 |
| HMDiMeCR | O | O | Cl | H | Me | 7.7 | 7.3 | 0.4 |
| HMJFR | O | O | F | H | Julolidine[b] | 9.8 | 9.8 | 0.0 |
| HMJCR | O | O | Cl | H | Julolidine[b] | 9.1 | 8.7 | 0.4 |
| HMSiR620h | SiMe$_2$ | O | H | H | Me | 5.0 | 4.8 | 0.2 |
| HMJSiR | SiMe$_2$ | O | H | H | Julolidine[b] | 6.6 | 7.1 | 0.4 |

Measured values are from the literature[16,24,27].
[a]Data were calculated with 5 explicit water molecules.
[b]In the case of $R^2$ = julolidine, the structure of the compound is as shown on the right.

(structure 1 in Supplementary Fig. 8, Supplementary data). We calculated p$K_{cycl}$ for these structures (Supplementary Fig. 9) and found that the calculated p$K_{cycl}$ values were in very good agreement with the measured values for most derivatives. We also tested another structure (structure 4 in Supplementary Fig. 8) and the structure with a four-water bridge, but the prediction was not improved (Supplementary Figs. 10 and 11).

These results are interesting, considering that the proton moves from the amino group to the hydroxymethyl group during the spirocyclization reaction. In order to evaluate the feasibility of this proton transfer, we searched for the transition state to evaluate the activation free energy of the ring-opening reaction, and conducted IRC calculation to identify the reaction path[23] (Supplementary Fig. 12). The activation free energy of the ring-opening reaction of HMTMR through this pair of structures ($C_A/O_A$−3water-1) was calculated to be 28.7 kJ mol$^{-1}$, which suggests that the reaction can proceed spontaneously at room temperature. These results imply that the $C_A/O_A$−3water-1 structures play an important role in the spirocyclization reaction of HMR derivatives. We also calculated various other structures produced by randomly arranging water molecules and found no hydrated structure that had a greater effect than this structure (Supplementary Fig. 13).

**Test of the calculation model**. In order to examine the versatility of our calculation method with a 3-water bridge, we next examined whether the p$K_{cycl}$ values of other HMR derivatives can be predicted by this method. We calculated p$K_{cycl}$ values of various HMR derivatives, including silicon-substituted derivatives at position 10[16], ring-fused derivatives[24], and asymmetric derivatives[24], together with those of aminomethyl rhodamine (AMR) derivatives[16]. We found that the calculated p$K_{cycl}$ values were in good agreement with the experimentally measured p$K_{cycl}$ values for almost all derivatives tested with exceptions of HMRG and

AMRG (Table 1). We speculated that additional explicit water molecule(s) might be required for predicting p$K_{cycl}$ of HMRG and AMRG, due to the presence of the non-substituted amino group of NH$_2$ at the xanthene ring. Thus, we calculated the hydration energy with four-water molecules and found that, in addition to the bridge of 3-water molecules, hydration at the amino group contributed to stabilization of the closed form (Supplementary Fig. 14), and this hydration can occur only when the amino group has an N–H bond. By including two additional water molecules, we succeeded in reproducing the measured p$K_{cycl}$ of HMRG and AMRG by calculation (Table 1). It is also necessary to consider hydration of the N–H bond of the amide group for predicting p$K_{cycl}$ of acetylated HMR derivatives (Supplementary Table 2, Supplementary data). We also tested the accuracy of the calculation method. We calculated p$K_{cycl}$ including a correction for dispersive interactions (Grimme's correction[25]), but this resulted in little improvement (Supplementary Table 3), and in some derivatives the bridge of 3-water molecules was difficult to optimize. We also tried a coarser calculation method, HF/6–31G*, but found that the bridge of 3-water molecules was not successfully optimized. Therefore, we concluded that the default method is the most suitable for p$K_{cycl}$ calculation.

In summary, our simple strategy of calculating the free energy difference for a structure including a 3-water crosslink between the xanthene ring amino group and the benzene ring hydroxymethyl group is sufficiently versatile to predict p$K_{cycl}$ of a wide variety of HMR derivatives without an N–H bond. Further, by adding explicit water molecules in an appropriate manner, our equilibrium model and calculation formula can be extended to predict p$K_{cycl}$ of HMR derivatives with an N–H bond.

**Molecular design based on calculational prediction**. As a next step, we applied the calculation model to predict the p$K_{cycl}$ values of HMR derivatives that have never been synthesized in order to

explore the correlation between structure and p$K_{cycl}$. First, we comprehensively calculated the p$K_{cycl}$ values of derivatives with a functional group introduced at the benzene moiety (Supplementary Fig. 15 and Supplementary Table 4). Electron-donating/withdrawing effects appeared at any position, but we found that introducing a substituent at position-3′ has a large p$K_{cycl}$-lowering effect, regardless of the electronic character of the group. For example, the calculated p$K_{cycl}$ of HMTMR is 9.5, while that of the 3′-methyl derivative (Supplementary Fig. 15) is 7.3. The effect is weaker in the case of substitution with F, which has a small radius, so there seems to be a tendency for the calculated p$K_{cycl}$ to become smaller with increasing bulk of the substituent at position-3′.

To confirm and investigate this substituent effect, we synthesized some 3′-substituted HMRG derivatives and measured their p$K_{cycl}$ values (Supplementary Figs. 16–19). As shown in Table 2, the measured values and the calculated values are in good agreement. Interestingly, in the case of 5MHMRG, the methyl group had no influence on p$K_{cycl}$. Based on these results, we supposed that the position-3′ effect is due to steric interaction between the substituent at position-3′ and the hydroxymethyl group at the adjacent position. In the open form, there is a one degree of freedom corresponding to dihedral rotation of the hydroxymethyl group, which contributes to the increase of entropy. On the other hand, the closed form is constrained because of the spiro-ring structure. Therefore, a bulky substituent at position-3′ decreases the entropy around the hydroxymethyl group only in the open form, which destabilizes the open form relative to the closed form and decreases p$K_{cycl}$. Indeed, substitution at position-3′ destabilized the open form (Supplementary Table 5). This finding that the bulk of the substituent at position-3′ is important, rather than its electronic character, should enable us to design fluorescent probes recognizing unprecedented targets that might be inaccessible with other methodologies for controlling fluorescence. Moreover, this effect can be used in probe design as a strategy to downwardly adjust p$K_{cycl}$.

Next, we found that distortion of the condensed spiro-ring leads to relative destabilization of the closed form and an increase of p$K_{cycl}$. For example, p$K_{cycl}$ calculation predicted that HMR derivatives in which benzene is replaced with a five-membered ring tend to have higher p$K_{cycl}$ values because of the strain effect in the spiro-ring (Supplementary Table 6). This effect can be used in probe design as a strategy to upwardly modify p$K_{cycl}$. Thus, by controlling the entropy and the strain effect around the hydroxymethyl group, we can rationally adjust p$K_{cycl}$ simply by structural modification of the benzene moiety. As already mentioned, it would be difficult to quantitatively predict these effects from empirical considerations, and this highlights the value of our non-empirical method.

**Design of fluorogenic probes based on p$K_{cycl}$ prediction.** The ability to carry out simultaneous multicolor imaging of different targets is a major advantage of optical fluorescence imaging. One of the goals of our research program is to develop a series of fluorogenic probes with different colors, targeting different enzymatic activities, in order to realize more precise and sensitive detection of tumors. We have already succeeded in developing a green fluorescent probe for aminopeptidase based on the HMR scaffold[18,26], obtaining an activation ratio of more than 800 upon reaction with the enzyme to form the highly fluorescent product HMRG, whose quantum yield is 0.81. We have been trying to design and develop yellow and red fluorogenic probes for aminopeptidases based on intramolecular spirocyclization, but we have not yet obtained satisfactory activation of fluorescence upon

**Table 2 Calculated and measured p$K_{cycl}$ values of HMRG derivatives substituted with various groups on the benzene moiety.**

| | Measured p$K_{cycl}$ | Calculated p$K_{cycl}$ | | | |
|---|---|---|---|---|---|
| X | 3′ (5′) | 3′ | 4′ | 5′ | 6′ |
| F | 8.0 | 8.1 | 8.3 | 7.4 | 8.1 |
| Me | 6.6 (8.2) | 6.1 | 8.1 | 7.9 | 7.2 |
| CF$_3$ | 5.3 | 5.3 | 7.1 | 7.4 | 7.6 |
| H | 8.1 | 7.9 | | | |

reaction with the target enzyme due to the difficulty of adjustment of p$K_{cycl}$, as well as the dimness of the fluorophores owing to their asymmetric structure[24,27]. Therefore, we next aimed to apply our present quantum chemical calculation methodology to design highly activatable yellow and red fluorogenic probes based on symmetrical position-10-substituted rhodamine derivatives such as SiR600 and CR550.

It is known that position-10-substituted HMR derivatives tend to have far lower p$K_{cycl}$ values than standard HMR derivatives with an oxygen atom at position 10. For example, p$K_{cycl}$ of HMSiR600 is 4.4 while that of HMRG is 8.1[27] (Fig. 3). In order to create fluorogenic probes that yield products with strong fluorescence intensity under neutral pH conditions, it is essential to find position-10-substituted HMR derivatives that have higher p$K_{cycl}$ values while retaining symmetrical structures with both of their amino groups in the form of NH$_2$ in order to secure a high fluorescence quantum yield. The conventional strategy to upwardly modify p$K_{cycl}$ within this structural limitation would be to weaken the nucleophilicity of the hydroxymethyl group by replacing it with some other nucleophilic functional group. However, the resulting effect on p$K_{cycl}$ is too large, and it is very difficult to obtain an appropriate p$K_{cycl}$ value.

Applying our new strategy, we calculated the p$K_{cycl}$ of various Si or C-substituted HMR derivatives having a five-membered ring in place of benzene. We looked for a derivative whose p$K_{cycl}$ is more than 8.5 without amino acid and less than 5.5 with amino acid or Ac group as an approximation. Among many Si- or C-substituted HMR derivatives, we found candidates with suitable p$K_{cycl}$ values for developing a GGT-activatable probe based on symmetrical fluorophores SiR600 and CR550 (Fig. 4). Taking into account ease of synthesis, we selected HMRR and HMRY (hydroxymethyl rhodamine rouge/yellow) and synthesized them to examine their spirocyclization properties (Supplementary Figs. 20 and 21). The measured values of p$K_{cycl}$ of HMRR and HMRY are 8.4 and 9.2 respectively, which are in very good agreement with the predicted values of 8.6 and 9.0, respectively. These results suggested that GGT-activatable probes could be created based on these structures (Fig. 5). Finally, to validate this expectation, we synthesized red and yellow fluorescent GGT probes, gGlu-HMRR and gGlu-HMRY, respectively, and confirmed that they are highly activated (>500- and >200-fold, respectively) by GGT (Supplementary Figs. 22 and 23). The former activation is much larger than that obtained with our previously developed red GGT probes[24,27] (Supplementary Table 7). We also confirmed that gGlu-HMRR and gGlu-HMRY function effectively as GGT

**Fig. 3 The influence of position-10 substitution on pKcycl and probe design.** Substitution of oxygen at position 10 by Si or other atoms is known to lower the pKcycl.

**Fig. 4 Search for optimal structures for red and yellow peptidase probe by pKcycl prediction.** The optimal derivative would have pKcycl < 5.5 when acetylated but pKcycl > after hydrolysis.

probes in a mouse model and can visualize tiny tumors in vivo (Fig. 5, Supplementary Figs. 24 and 25). Thus, our approach enabled us to design fluorogenic probes with pinpoint accuracy without the need to synthesize multiple reference compounds; this would not have been possible using conventional design strategies.

In conclusion, we scrutinized the intramolecular spirocyclization of HMR derivatives and found that consideration of explicit water molecules is essential for accurately estimating the free energy difference between the closed and open forms. Our calculations well reproduced the pKcycl values of known HMR derivatives and could predict those of unknown derivatives. This method of pKcycl prediction can be applied to a variety of HMR derivatives, and we believe it has the potential to enable the design of new fluorogenic probes targeting unprecedented biochemical reactions without the need for tedious synthesis of multiple reference or pilot compounds. As proof of concept, we applied this pKcycl prediction methodology to design yellow and red GGT-activatable fluorogenic probes with very high activation

ratios and confirmed their ability to visualize tiny tumors in a mouse model.

The present methodology is quite simple and based on calculations of the ground-state equilibrium, so a similar approach should be applicable for designing compounds with not only specified fluorescence characteristics, but also specified photosensitization properties, photoacoustic spectra, uncaging behavior and photo-labeling reactions. We anticipate that control of the thermodynamic equilibrium of molecules based on quantum chemical calculations will be an important strategy in future molecular design.

## Methods

**Computational details**. We performed calculations using the Gaussian09 program[28]. General geometry optimization and vibrational analysis of local minima and transition states and IRC calculation were performed at the B3LYP/6–31G(d) level including water in the PCM model. Stationary points were optimized without any symmetry assumptions or correction of dispersive interactions, and we used tight convergence criteria.

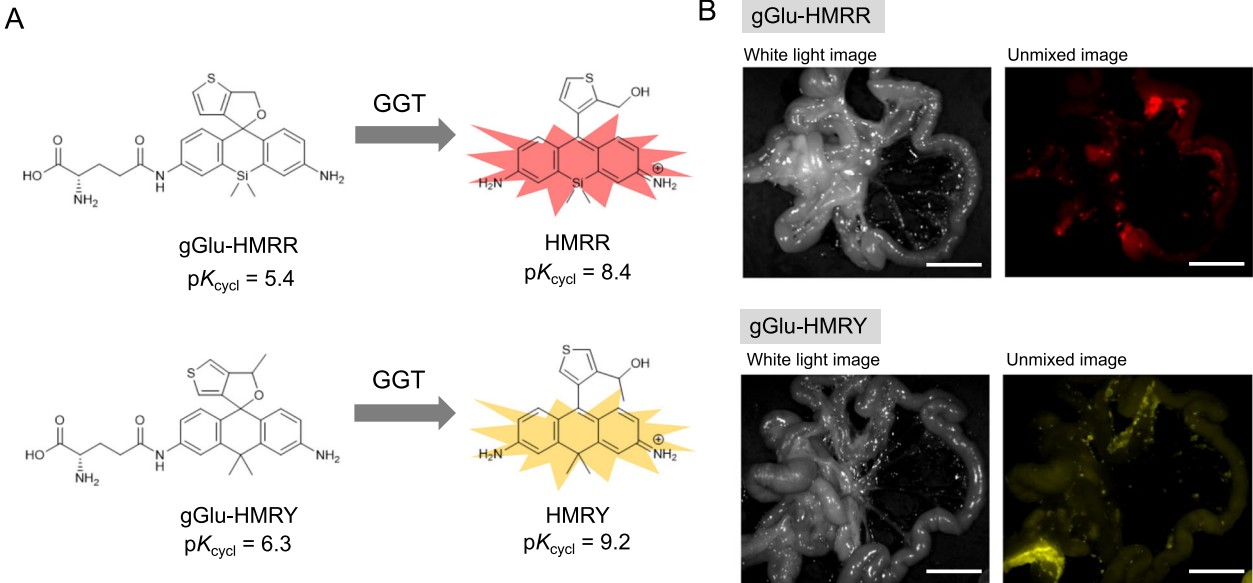

**Fig. 5 Design of fluorogenic probes based on p$K_{cycl}$ prediction. a** Measured p$K_{cycl}$ of computationally designed derivatives and red and yellow GGT probes based on them. **b** Fluorescence spectral imaging of mouse models of peritoneal metastases at 5 min post treatment with probes (100 µM, 300 µL). Excitation, 575–605 nm (HMRR), 503–555 nm (HMRY); emission, 645 nm long pass (HMRR), 580 nm long pass (HMRY). Scale bar: 1 cm.

**Tumor model of peritoneal implants**. All procedures were carried out in compliance with the Guide for the Care and Use of Laboratory Animal Resources and the National Research Council and were approved by the Institutional Animal Care and Use Committee. The tumor implants were established by intraperitoneal injection of $1 \times 10^6$ SHIN3 cells[29] suspended in 300 ml of PBS into 7-week-old female nude mice (CLEA Japan, Inc., Japan). Experiments with tumor-bearing mice were performed at 30 days after injection of SHIN3 cells.

**In vivo spectral fluorescence imaging**. Mice were injected intraperitoneally with 300 µl of 100 µM probe solution. After 5 min, the mice were killed with isoflurane and the abdominal cavity was exposed. Fluorescence images were obtained with the Maestro In-Vivo imaging system (CRi Inc.). The yellow-filter setting (excitation, 575–605 nm; emission, 645 nm long pass) and green-filter setting (excitation, 503–555 nm; emission, 580 nm long pass) were used. The tunable filter was automatically stepped in 10-nm increments, from 500 to 800 nm, while the camera sequentially captured images at each wavelength interval. The spectral fluorescence images consisting of autofluorescence and HMRG spectra were unmixed for visual assessments with Maestro software.

## Data availability
The datasets of the current study are available from the corresponding author on reasonable request.

## Code availability
Code for randomized calculation (Supplementary Fig. 13) is available in Github (https://github.com/TachibanaRyo-moroba/Molecular-Design-Strategy-HMR/). System requirements: Python 3.7 with additional modules (pandas, numpy, network, matplotlib).

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

## Acknowledgements

Calculations were performed on the supercomputer of ACCMS, Kyoto University. SHIN3 cells were obtained from Dr. Hisataka Kobayashi, NCI, NIH. This research was supported in part by AMED/CREST;JP19gm0710008 (to Y.U.), by JST/PRESTO (JPMJPR14F8 to M.K.), by MEXT/JSPS KAKENHI; JP16H02606, JP26111012 and JP19H05632 (to Y.U.), JP15H05951'Resonance Bio' (to M.K.), 17H01387 (to S.S.), 15H02158 (to K.M.), by JSPS Core-to-Core Program, A. Advanced Research Networks, by a grant from Hoansha Foundation (to Y.U.), by Japan foundation for applied enzymology (to M.K.).

## Author contributions

R.T. and A.N. conducted experiments, performed analyses, and wrote the paper. S.S. and K. M. supervised the project. M.K and Y.U. planned and initiated the project, designed and conducted experiments, wrote the paper, and supervised the entire project.

## Competing interests

The authors declare no competing interests.
