## [Peer Review File · Communications Chemistry]

Reviewers' comments:

Reviewer #1 (Remarks to the Author):

Urano and co-workers developed a theoretical approach to predict the pK_{cycl} of HMR derivatives using computational chemistry, based on which fluorogenic rhodamine probes can be rationally designed without experimental screening. This approach would greatly speed up the development of rhodamine probes with tailed spirocyclization reactions for various biosensing applications. As a proof of concept, they also computationally designed and experimentally validated one red and one yellow GGT fluorogenic probes. These results are expected to have a profound impact on the fluorescence probe community, especially those based on various rhodamine fluorophores.

The reviewer suggests a minor revision to this manuscript by addressing several issues on the computational methods and format. Detailed comments are attached below.

1. Computational methods: a few water molecules are added to rhodamines in the reported calculations. In such cases, the dispersive interactions between molecules should be taken into consideration for the accurate modeling of intermolecular interactions, and the resulted Gibbs free energy.

This correction, i.e., Grimme's correction, is necessary for B3LYP (B3LYP-D3 or D4). See: J. Chem. Phys., 2010, 132, 154104; J. Phys. Chem. Lett. 2016, 7, 2197-2203 for more info.

However, in the computational methods, the authors did not mention these details, and it is not clear if such corrections are made.

2. Lines 105-106, the authors should briefly explain the reasons why the concentration of OB can be ignored.

3. Lines 141-145, the authors suggested that IRC calculations have been performed to confirm the reaction path. Such data is key for understanding the ring-opening reactions and should be included in the SI.

4. References are outdated. The latest references are up to Year 2018 only. The authors should cite the latest references, especially those on rhodamine probes and spirocyclization equilibrium mechanism.

5. Supporting figures should be cited and appear in sequence whenever possible. In this case, Figures 1, 7, 10 etc. are not mentioned in the main text. This could be easily addressed. A similar check should also be done for other supporting figures and tables.

6. The format of the SI could be further improved. For example, "Table S 2 Calculated pK_{cycl} values of HMRG, AMRG, HMAcRG with first-shell water molecules (3 water bridge and additional molecules)". An extra space appears between "S" and "2". The caption head "Table S2" and text "Calculated..." should be joined together, instead of appearing in two lines. Similar changes should also be made to other figure/table captions.

7. Typos. "(Figure S6, S7)" should be "(Figures S6 and S7)".

Reviewer #2 (Remarks to the Author):

The manuscript is publishable after revision in the following directions.

1. The authors do not mention the level of the theory they have used in the optimization of the geometry of the molecules they have studied.

2, The authors need to provide the Cartesian coordinates of the optimized geometries for quick verification.

Reviewer #3 (Remarks to the Author):

In this paper, the authors describe the rational design strategy of fluorescent probe by utilizing quantum chemical prediction of intramolecular spirocyclization. Fluorogenic probes are useful tools for biochemical analysis. To design fluorogenic probes rationally, several design strategies have been developed such as the control of photoinduced electron transfer (PeT) and Förster resonance energy transfer (FRET). The authors newly proposed the design strategy of fluorescent probe based on rational control of intramolecular spirocyclization. They studied intramolecular spirocyclization of their compounds and succeeded in the pKcycl prediction method by using estimation of the free energy difference between the closed and open forms with explicit water. By utilizing their pKcycl prediction method, they finally demonstrated the development of two high-performance fluorescence probes for GGT and applied visualization of tiny tumors in mouse model. The data such as the calculation of pKcycle, measured pKcycle values, and the fluorescence properties of new probes were well characterized and described. In addition, this approach is interesting and can be evaluated because this approach enabled us to easily design fluorogenic probes based on intramolecular spirocyclization mechanism. Therefore, I recommend publication of this manuscript in Communications Chemistry after minor revisions.

The following comments should be considered.

- 1) In Figure S21, the experimental conditions such as detection wavelength are not described, so they should be described. And it is useful for readers to show the fluorescence spectrum change in the reaction with GGT.
- 2) Regarding Figure 3A, Si in the chemical formula is small and hard to see.
- 3) reference S1 is incomplete.

Point-by-point response to comments of reviewers

Manuscript ID: COMMSCHEM-20-0049

TITLE: Molecular Design Strategy of Fluorogenic Probes Based on Quantum Chemical Prediction of Intramolecular Spirocyclization

In order to address the reviewers' comments, we have revised the text and carried out additional experiments. Each comment from the reviewers is reproduced below, followed by our reply. Changes in the manuscript and supporting information are highlighted in yellow.

Responses to Reviewer 1

Reviewer #1 (Remarks to the Author):

Urano and co-workers developed a theoretical approach to predict the pK_{cycl} of HMR derivatives using computational chemistry, based on which fluorogenic rhodamine probes can be rationally designed without experimental screening. This approach would greatly speed up the development of rhodamine probes with tailed spirocyclization reactions for various biosensing applications. As a proof of concept, they also computationally designed and experimentally validated one red and one yellow GGT fluorogenic probes. These results are expected to have a profound impact on the fluorescence probe community, especially those based on various rhodamine fluorophores.

The reviewer suggests a minor revision to this manuscript by addressing several issues on the computational methods and format. Detailed comments are attached below.

1. Computational methods: a few water molecules are added to rhodamines in the reported calculations. In such cases, the dispersive interactions between molecules should be taken into consideration for the accurate modeling of intermolecular interactions, and the resulted Gibbs free energy.

This correction, i.e., Grimme's correction, is necessary for B3LYP (B3LYP-D3 or D4). See: J. Chem. Phys., 2010, 132, 154104; J. Phys. Chem. Lett. 2016, 7, 2197-2203 for more info.

However, in the computational methods, the authors did not mention these details, and it is not clear if such corrections are made.

R1-1 Thank you for pointing this out. In the previously submitted manuscript, we did not include

the correction of dispersive interactions. So we newly calculated pK_{cycl} with Grimme's correction and compared the results with the previous ones (Supplementary Table 3 and 4). It's not clear why, but we found little improvement in the prediction of pK_{cycl} values when the correction of dispersive interactions was included. Considering the cost of calculation, we concluded that the default condition is the most suitable for the prediction of pK_{cycl} . To explain these results, we added some sentences in the "Test of the calculation model" section in the main text.

2. Lines 105-106, the authors should briefly explain the reasons why the concentration of OB can be ignored.

R1-2 We added a comment in the main text. In the case of aminomethyl derivatives, the K_{aOH} value is very small ($K_{\text{aOH}} < 10^{-30}$).

3. Lines 141-145, the authors suggested that IRC calculations have been performed to confirm the reaction path. Such data is key for understanding the ring-opening reactions and should be included in the SI.

R1-3 We agree that the IRC calculations are important for understanding the ring-opening reactions, and have added the data in the supplementary information (Supplementary Figure 12) as suggested. Under the tested condition, the proton of the amino group in the closed form took a stable structure at the center of the bridge in the process of proton transfer to the spiro ring. We haven't found a transition state from structure 3 to O_A -3water-1 (rotation of the hydroxymethyl group), but since these structures have similar energy values ($\Delta E = 0.8$ kJ/mol), we think that transition between them would be fast.

4. References are outdated. The latest references are up to Year 2018 only. The authors should cite the latest references, especially those on rhodamine probes and spirocyclization equilibrium mechanism.

R1-4 We added some later references.

5. Supporting figures should be cited and appear in sequence whenever possible. In this case, Figures 1, 7, 10 etc. are not mentioned in the main text. This could be easily addressed. A similar check should also be done for other supporting figures and tables.

R1-5 We have made sure all supplementary figures and tables are mentioned in the main text.

6. The format of the SI could be further improved. For example, “Table S 2 Calculated pK_{cycl} values of HMRG, AMRG, HMAcRG with first-shell water molecules (3 water bridge and additional molecules)”. An extra space appears between “S” and “2”. The caption head “Table S2” and text “Calculated...” should be joined together, instead of appearing in two lines. Similar changes should also be made to other figure/table captions.

R1-6 We checked all the captions and formats, and modified them to conform to the journal style.

7. Typos. “(Figure S6, S7)” should be “(Figures S6 and S7)”.

R1-7 We corrected it. Thank you for your check.

Responses to Reviewer 2

The manuscript is publishable after revision in the following directions.

1. The authors do not mention the level of the theory they have used in the optimization of the geometry of the molecules they have studied.

R2-1 Thank you for this important comment. We added descriptions of the level of the theory in the methods section in the main text, according to the style of this journal.

2, The authors need to provide the Cartesian coordinates of the optimized geometries for quick verification.

R2-2 Thank you. We added the Cartesian coordinates of the optimized geometries used in pK_{cycl} calculation in the supplementary information.

Responses to Reviewer 3

In this paper, the authors describe the rational design strategy of fluorescent probe by utilizing quantum chemical prediction of intramolecular spirocyclization. Fluorogenic probes are useful tools for biochemical analysis. To design fluorogenic probes rationally, several design strategies have been

developed such as the control of photoinduced electron transfer (PeT) and Förster resonance energy transfer (FRET). The authors newly proposed the design strategy of fluorescent probe based on rational control of intramolecular spirocyclization. They studied intramolecular spirocyclization of their compounds and succeeded in the pKcycl prediction method by using estimation of the free energy difference between the closed and open forms with explicit water. By utilizing their pKcycl prediction method, they finally demonstrated the development of two high-performance fluorescence probes for GGT and applied visualization of tiny tumors in mouse model. The data such as the calculation of

pKcycle, measured pKcycle values, and the fluorescence properties of new probes were well characterized and described. In addition, this approach is interesting and can be evaluated because this approach enabled us to easily design fluorogenic probes based on intramolecular spirocyclization mechanism. Therefore, I recommend publication of this manuscript in Communications Chemistry after minor revisions.

The following comments should be considered.

1) In Figure S21, the experimental conditions such as detection wavelength are not described, so they should be described.

And it is useful for readers to show the fluorescence spectrum change in the reaction with GGT.

R3-1 Thank you for these suggestions. We add the information about detection wavelength. We also conducted new in vitro experiments, and recorded the fluorescence spectrum change in the reaction of the newly developed probes with GGT; these results were added as figure S23.

2) Regarding Figure 3A, Si in the chemical formula is small and hard to see.

R3-2 We modified the figure. Thank you for your check.

3) reference S1 is incomplete.

R3-3 We corrected the reference. Thank you for your check.

REVIEWERS' COMMENTS:

Reviewer #1 (Remarks to the Author):

The authors have addressed all of my concerns. This manuscript is now suitable for publication.